# How Precisely Can Easily Accessible Variables Predict Achilles and Patellar Tendon Forces during Running?

**DOI:** 10.3390/s21217418

**Published:** 2021-11-08

**Authors:** René B. K. Brund, Rasmus Waagepetersen, Rasmus O. Nielsen, John Rasmussen, Michael S. Nielsen, Christian H. Andersen, Mark de Zee

**Affiliations:** 1Sport Sciences–Performance and Technology, Department of Health Science and Technology, Aalborg University, 9220 Aalborg, Denmark; skovsgaard_04@yahoo.dk (M.S.N.); christian_hauge_and@hotmail.com (C.H.A.); mdz@hst.aau.dk (M.d.Z.); 2Department of Mathematical Sciences, Aalborg University, 9220 Aalborg, Denmark; rw@math.aau.dk; 3Department of Public Health, Aarhus University, 8000 Aarhus, Denmark; roen@ph.au.dk; 4Research Unit for General Practice, 8000 Aarhus, Denmark; 5Department of Materials and Production, Aalborg University, 9220 Aarhus, Denmark; jr@mp.aau.dk

**Keywords:** Garmin, wearables, Achilles tendon, patellar tendon, algorithm, injuries, sports medicine

## Abstract

Patellar and Achilles tendinopathy commonly affect runners. Developing algorithms to predict cumulative force in these structures may help prevent these injuries. Importantly, such algorithms should be fueled with data that are easily accessible while completing a running session outside a biomechanical laboratory. Therefore, the main objective of this study was to investigate whether algorithms can be developed for predicting patellar and Achilles tendon force and impulse during running using measures that can be easily collected by runners using commercially available devices. A secondary objective was to evaluate the predictive performance of the algorithms against the commonly used running distance. Trials of 24 recreational runners were collected with an Xsens suit and a Garmin Forerunner 735XT at three different intended running speeds. Data were analyzed using a mixed-effects multiple regression model, which was used to model the association between the estimated forces in anatomical structures and the training load variables during the fixed running speeds. This provides twelve algorithms for predicting patellar or Achilles tendon peak force and impulse per stride. The algorithms developed in the current study were always superior to the running distance algorithm.

## 1. Introduction

Exercise should be taken seriously since it is believed to have profound health benefits [1]. One type of exercise activity is running, which, on a global scale, has gained popularity in the past decades. Running is preferred by many, owing to its accessibility and beneficial effects on various health-related outcomes, such as fitness level and health [2].

In contrast to its benefits, running can also lead to injuries in the musculoskeletal system [3]. Patellar and Achilles tendinopathy account for more than 10% of all running-related injuries [4]. These conditions and running-related injury, in general, are major obstacles to exercise activity [5], so prevention of patellar and Achilles tendon injuries are important. In-depth knowledge about forces applied to the involved anatomical structures is needed because an injury occurs when the cumulative tendon load exceeds the structure’s capacity to withstand the load [6,7,8]. Cumulative tendon load is considered a superior metric for the prediction of injury compared to running distance [9], which has been widely used in the previous literature [10]. Therefore, more sophisticated measures of load are warranted.

An advanced method to quantify load is to estimate tendon force using computational models of the musculoskeletal system [11]. Unfortunately, this method seems practically and logistically impossible in a real-life setting due to high computational complexity and demands of detailed motion data. Therefore, developing a computationally simple algorithm to predict the cumulative force in the Achilles or the patellar tendon is an important step to improving the understanding of the etiology underpinning running injury in these structures. If successful, such algorithms can be used to obtain session-specific and structure-specific approximations of tissue loads in large-scale epidemiological studies examining the “too much training load, too soon”-theory [7]. Large-scale studies are needed to assess changes in the tendon force in different groups displaying different recovery patterns [7], running experience [6], previous injuries [6], and pain sensitivity [12], to name a few. In the ideal study, thousands of runners should be included to consider various effect-measure modifiers. However, such a sample size is likely to make it difficult to obtain full-body kinematic and kinetic data in a time-efficient manner.

To obtain enough personalized data on individual runners, an algorithm predicting cumulative force on an Achilles or patellar tendon should rely on measures that are easy to assess in-situ, rather than advanced measures that can only be assessed in biomechanical laboratories. As an example, running measures (e.g., speed, cadence), which are measurable by commercially available devices, such as smartwatches, may be used when developing algorithms to predict step-specific forces in the patellar and Achilles tendons. Commercially available devices are widely used by runners, as they can be worn unobtrusively during running [13,14]. These devices have made it possible to obtain indirect measures of training load (such as the number of strides, cadence, ground contact time, and vertical oscillation) in an outdoor environment [15,16]. Using such measures to calculate approximations of forces in the patellar and Achilles tendon force requires, however, that the approximations have acceptable predictability of biomechanically-assessed forces [6,17].

Therefore, the main objective of the present study was to investigate whether an algorithm can be developed to estimate patellar and Achilles tendon forces and impulses per stride during running, using measures that can be easily collected by Garmin devices without expert assistance. A secondary objective was to evaluate the predictive performance of the algorithm while estimating patellar and Achilles tendon peak force and impulse per stride and compare the algorithm with running distance metrics, which is the common way in the literature of estimating cumulative load and serves as a control.

## 2. Materials and Methods

### 2.1. Subjects

Twenty-four runners (17 males and 7 females) were included in the study with an average age of 26 ± 1.3 years. The runners weighed 82 ± 11 kg; stature 182 ± 7 cm; and had a knee, ankle, and shoe sole height of 49.6 ± 3.1 cm, 7.6 ± 0.9 cm, and 3 ± 0.9 cm, respectively (see Appendix A for more detailed numbers). All runners were recreationally active for at least 60 min per week. Additionally, all runners had been injury-free for at least six months and completed the setup described in the protocol without any complaints, pain, or discomfort. Before testing, the runners were informed about the purpose of the study, study design, equipment, and signed a declaration of informed written consent. The Regional Ethics Committee of North Jutland waived the approval of the study owing to the study design (an observational cross-sectional study) since observational studies do not require approval from the local ethics committee according to Danish Law. 

### 2.2. Procedures

Prior to data collection, each runner was introduced to the Garmin Forerunner 735XT (GFR) to become familiar with the start, stop and save function. Anthropometric data for each runner were collected based on instructions provided by Xsens (Xsens Technologies B.V, Enschede, The Netherlands). Seventeen inertial motion units (IMU) were mounted on the Xsens suit on the following anatomical locations: head, sternum, pelvis, upper legs, lower legs, feet, shoulders, upper arms, forearms, and hands using the designated clothing items of the Xsens system. The anthropometric data were loaded into the Xsens MVN Studio 4.3 before calibration. Prior to the data collection, the runners performed a 20-min warm-up at a self-selected pace, followed by five-min rest. During this resting period, a calibration of the inertial motion capture system was performed. Segment orientations were obtained by applying the IMU-to-segment alignment, found using a known upright pose (N-pose) [18]. This N-pose calibration updates the joints and external contacts to limit the position drift [18]. If the calibration was categorized as “good” according to the system, it was accepted and redone otherwise. A visual inspection using the live view of the joint movement was performed to ensure that the recorded data were consistent with the movement of the runner. The calibration was performed outdoor to reduce the amount of magnetic disturbance [18]. Roetenberg, Luinge, and Slycke provide a detailed description of the Xsens system [18]. The runner was equipped with the GFR after the calibration was completed. 

### 2.3. Data Collection

Each of the runners performed three running trials of two minutes on a straight outdoor track paved with asphalt at three different speeds (10, 12, and 14 km/h) in randomized order. The speed was controlled by a person riding a bike in front of the runner with a Garmin Fenix 2 GPS watch (Garmin Ltd., Olathe, KS, USA) mounted on the bike, while another rode behind the runner on a Long John bicycle with a computer, a battery and access point for data collection. The Xsens system was activated first. Then, the runner was instructed to perform a jump and start the GFR upon landing to synchronize the two datasets. 3D kinematic data of the full-body were recorded at 240 Hz with an Xsens MVN link motion capture suit. Running dynamic data were recorded each second (60 Hz) with the GFR paired with a heart rate strap (HRM-Run; Garmin Ltd., Olathe, KS, USA). The GFR variables that were measured during the run and included in the present study were: instantaneous Speed, Vertical Oscillation, Ground Contact Time, Step length, and Cadence.

### 2.4. Data Processing

Data from GFR were downloaded using Garmin Connect software 7.1.4.0 (Garmin Ltd., Olathe, KS, USA) and exported to R (v. 4.0.5). The Xsens data was captured in a native file format called MVN), which was HD reprocessed in MVN studio to get the best performance for the recorded motion. The data were aligned using the jump as an indicator of the start of the Garmin data. Furthermore, 35 s of data were disregarded from the start of both systems to make sure the runner had reached a constant speed.

The kinematic data from Xsens were exported in Biovision Hierarchy (BVH) format and thereafter processed in a computer model (BVH_Xsens template in AnyBody Managed Model Repository version 2.2) of the musculoskeletal system using the AnyBody Modeling System (version 7.2). In the AnyBody Modeling System, the patellar and Achilles tendon forces were estimated for four strides per trial [19,20,21,22], using the muscle recruitment approach described in Damsgaard et al. [21]. We predicted ground reaction forces from the kinematics using the method described by Skals et al. [20]. Ground reaction forces were predicted by creating 25 contact points under each foot of the musculoskeletal model. Each contact point consisted of five unilateral force actuators, which could generate a positive vertical force orthogonal to the ground, and static friction forces in the two horizontal directions using a friction coefficient of 0.8. In addition, to compensate for the sole thickness of the running shoes, a 25 mm height activation offset threshold was added to the musculoskeletal model.

Each runner provided 24 running strides, giving 576 running strides in total (=24 runners × 4 strides × 2 legs × 3 speeds) with an estimate of patellar and Achilles tendon force. Each of the four strides was paired with the observation from Garmin closest in time to this measure.

### 2.5. Statistics

Data were analyzed using a mixed-effects multiple regression model, which was used to model the association between the estimated forces in anatomical structures and the training load variables during the fixed running speeds. In the mixed model, runner-specific random effects are used to account for possible correlations between repeated observations for each runner at different running speeds.

The response variable is either the estimated peak force or impulse per stride in either the patellar tendon or Achilles tendon. The predictor variables are the different training load variables from Garmin (speed, ground contact time, vertical oscillation, and cadence) and anthropometric variables of the runners, including body mass (kilogram), sex, knee height (centimeter), ankle height (centimeter), shoe sole height (centimeter) and body height (centimeter) or the traditionally used running distance (approximated by distance per stride). Once a model is established for the relation between a response variable and the predictor variables, an algorithm for predicting the response variable is directly obtained from the model prediction equation.

For each of the four response variables, three models were obtained. The first model is based on the “distance” as input, which is the common way in the literature of estimating cumulative load and serves as a control [10]. The second model is the “best fitted”, which considers all available predictors and serves as a benchmark for the predictive performance and should be used when possible (See Table 1 and Table 2). The third model is the “practically feasible” which considers only predictors, which may be assessed validly by runners themselves. 

The reason for fitting a best-fitted and practically feasible model stems from our experience from previous studies [7,17,22]. Some anthropometric measurements are difficult to collect in a large-scale project (ankle and knee height), in contrast to other measures such as running measures, sex, and body height, which may be assessed validly by runners themselves. Hence, for the best-fitted model, all 10 variables were available, while the practically feasible disregarded knee, ankle, and shoe sole height and the distance model only used the distance per stride. 

The best fitted, as well as the practically feasible models, were identified by fitting all potential combinations of predictors (best fitted model:2^10^ = 1024 and practically feasible model:2^6^ = 64). For each response variable, the combination of predictors that minimized prediction error was selected. The prediction error (*PE*; Equation (1)) was computed for each combination of predictors using a 5-fold cross-validation approach [23,24] comparing the difference between the tendon force estimated using AnyBody and the tendon force predicted by the algorithm given by the considered combination of predictors.
(1)$$PE=\sqrt{{\left[observed-predicted\right]}^{2}\;}$$

A relative proportion of prediction error (*PPE*; Equation (2)) was developed to get an impression of the size of error within and between the structure-specific forces: (2)$$PPE=\frac{prediction\;error\;\left(N\right)}{mean\;structurespecific\;\left(N\right)}100$$

The statistical analyses were performed in R.

The *PE* and *PPE* were also used to compare the best fitted and practically feasible models with the distance-based model. In addition, we estimated the Pseudo R-Squared value based on the approach described by Nakagawa and Schielzeth [25].

## 3. Results

Garmin data for two runners could not be used owing to an error in the acquisition of data (Zeroes in the stride length and unrealistically low vertical oscillation). For one of these runners, the error affected the Garmin data for the 10 km/h speed and, for the other, the error affected all three speeds (see Figure 1). For six runners, the estimated ground reaction forces from AnyBody were unrealistically high on their right leg on either 10, 12, or 14 km/h (see Figure 1). After removing the noisy recordings, we ended up with 520 eligible strides in total (10.7% trials were lost in the process) from 23 different runners. On average, the 23 runners displayed a peak patellar force of 5268 ± 915 N, an impulse per stride of 152.7 ± 33.3 kNs, while the Achilles tendon peak force was 5150 ± 1500 N and impulse per stride was 112.4 ± 40.6 kNs (See Appendix A for the exact values). 

### 3.1. Speeds Association with Step-Specific Tendon Force

Figure 2 shows the association between different speeds and loads on anatomical structures, training load, and anthropometrics. Achilles (slope: 309; *p*-value: <0.001) and patellar (slope: 59; *p*-value: 0.001) tendon peak forces were positively affected by increased running speed. Achilles tendon (slope: −920; *p*-value: 0.305) impulse per stride was negatively but insignificantly related to running speed.

Patellar (slope: −5928; *p*-value: <0.001) peak forces were negatively affected by running speed. Stride length (slope: 82; *p*-value: <0.001), vertical oscillation (slope: 1.66; *p*-value: <0.001) and cadence (slope: 1.08; *p*-value: <0.001) were positively associated with increased running speed, while ground contact time (slope: −10; *p*-value: <0.001) was negatively associated with increased running speed. Moreover, the exact values for the intended running speeds are descripted in Appendix A).

### 3.2. The Algorithms

For each of the four force variables, the best fitted and practically feasible models were selected according to the PE defined in equation 1 (See Figure 3 and Figure 4 and Table 1 and Table 2) resulting in best fitted and practically feasible algorithms for predicting each of the four force variables.

With regards to peak patellar tendon force, speed, cadence, body mass, sex, and body height were chosen for both the best fitted and the practically feasible algorithms (see Table 1). The signs of the coefficients for these predictors were consistent between the two algorithms. Moreover, for the algorithms predicting patellar tendon impulse per stride, speed, ground contact time, vertical oscillation, cadence, sex, and body height were selected for both algorithms and with consistent coefficient signs. Moreover, knee, ankle, and shoe sole height were selected in the best-fitted algorithms for both the peak patellar tendon force and impulse per stride.

Speed and body height were selected in both algorithms for peak Achilles tendon force with the same signs and similar coefficients (see Table 1). For Achilles tendon impulse per stride, vertical oscillation, body mass, and body height were selected in both algorithms, again with the same signs (see Table 2). Moreover, ankle and shoe sole height were selected in the best-fitted algorithms both for the peak Achilles tendon force and impulse. Equation (3) demonstrates how the algorithms can be used to predict the patellar tendon peak force (*PTF*) based on the practically feasible algorithm:(3)PTF≈1615+141Speed−46.56Cadence+10Bodyweight+484Male+90Body height  

The performance of the algorithms was evaluated as the standard deviation between and within runners and the absolute accuracy (standard deviation of the prediction error) computed by cross-validation. The standard deviation (SD) within runners remained the same in both approaches, while the SD between runners were larger for the practically feasible algorithms (See Figure 3 and Figure 4). This is also in agreement with PE and PPE, which increases with a similar trend as the between SD for best fitted and practically feasible algorithm. Since knee, ankle, and shoe sole height are not included in the practically feasible algorithms, force variation explained by these variables in the best-fitted models is instead considered to be random runner-specific variation in the practically feasible models. This explains the higher between SDs for the practically feasible algorithms.

### 3.3. Predictive Performance

The proportion of prediction error was 16% or below for all patellar tendon algorithms and 34% or below for all Achilles tendon algorithms. The algorithms fitted using distance per stride revealed, for the patellar tendon, a prediction error of 934 N for peak force and 35.1 kNs for impulse, while the proportions of prediction error were 18% and 23%, respectively. For the Achilles tendon, the prediction error was 1532 N for peak force and 41.0 kNs for impulse, giving proportions of the prediction error of 30% and 36%, respectively. A graphical comparison of the different algorithms is provided in Figure 3 and Figure 4, with the predicted structure-specific force plotted against the AnyBody-estimated force. From a visual inspection of the figures, it is evident that the scatters fall closer to the identity black line for the best fitted and practically feasible algorithms, compared to the distance-based prediction. The Pseudo R-squared value of the fixed effects improved for both algorithms compared to the Distance algorithm (see Table 1 and Table 2). 

## 4. Discussion

The purpose of this study was to explore the predictive performance of algorithms to predict patellar and Achilles tendon force. The algorithms were based on self-reported data from runners and data from a Garmin running device quantifying running measures while running in an outdoor environment.

### 4.1. Predictive Performance

For both the patellar tendon and Achilles tendon, the prediction error and proportion of prediction error were greatest for the distance algorithm and smallest for the best fitting algorithm. This could indicate that, when possible, the best fitting algorithm may be preferred over the practically feasible algorithm. Moreover, adding anthropometric information to the training load variables measured from the Garmin running device improved the accuracy of all algorithms. This indicates that anthropometric measurements should be used when feasible.

### 4.2. Comparing Structure Specific Forces

Using AnyBody to Predict ground reaction force has previously been demonstrated to yield valid estimates [19,20]. Different musculoskeletal models have also demonstrated agreement on changes of structural forces at different movement speeds.

Moreover, the estimated structural forces are similar to forces in other studies. The patellar tendon peak force increased from 10 km/h to 12 km/h, while no increase was found with the increase from 12 km/h to 14 km/h. A similar study mimicking the patellar tendon force by applying the peak muscle activity in the vasti muscles, demonstrated a similar pattern, although the muscle activity decreased from 18.6 km/h to 25 km/h [26]. Moreover, using a force transducer on the Achilles tendon, Kharazi et al. [27] demonstrated that the Achilles tendon peak force grows linearly from 5–12.6 km/h. With a similar method, Komi [28] demonstrated that the Achilles tendon peak force was increasing until around 21 km/h (6 m/s) and thereafter decreasing at higher speeds. They directly measured the forces in the Achilles tendon under local anesthetization and found similar forces as in the present study. At 10.8 km/h (3 m/s) the force was around 5 kN, while we found that, at 10 and 12 km/h, it was on average 4.86 kN and 5.16 kN, respectively. At 14.4 km/h (5 m/s) they found the force to be around 6 kN which compares well with the 5.7 kN that we found at 14 km/h.

### 4.3. Limitations

The study design is limited to low-range running speeds from 10 km/h to 14 km/h and by the lack of precise running speed measurement. A plateau likely exists regarding vertical oscillation, ground contact time, stride length, and cadence at faster speeds. In the present study, a linear increase was found for stride length, ground contact time, and cadence, while a slightly progressive increase in vertical oscillation was found. However, studies including higher speed have demonstrated that stride length increases at a lower rate when the speed increases [26]. Consequently, the opposite seems to be the case for cadence, which had progressively higher increases in cadence over time. However, Dorn et al. [26] demonstrated a linear reduction in ground contact time from 12.5–32 km/h which may cover the range of speed performed during endurance running. This may indicate that the present study was limited in the speed range and the algorithm should be used with caution for predicting structural forces below 10 km/h and above 14 km/h. Moreover, the dataset was rather small and it can be questioned whether the runners included are representative of all types of runners.

Since the data had rather large noise/random variance, the prediction algorithms may display reduced predictive performance. Several improvements have the potential to reduce this noise: Firstly, the Garmin device is not providing stride-to-stride data. Consequently, it was not possible to synchronize the stride-to-stride data from Xsens precisely with the stride-to-stride data from Garmin. Instead, we used second-to-second data from the Garmin device. Secondly, the measures we used are only an average of a series of previous steps, which together with the lack of stride data may lead to imprecise predictions of the tissue loading. Stride-to-stride data without averages of previous steps will improve the fitting of the algorithm. This will ensure an appropriate synchronization between data. Thirdly, another way of reducing the noise could be to change the prediction of ground reaction forces from full-body kinematics to measuring ground reaction forces. Finally, control of running speed could be improved by better measurement techniques, and this might reduce random noise and enhance the significance of the findings. Still, the algorithms may prove beneficial in their current form, as the prediction error of the algorithms were at least 100 N below the prediction error derived from using the running distance algorithm, which is the commonly used exposure metric in previous literature [10].

### 4.4. Perspectives

Running injuries are commonly affecting runners and can lead to a temporary or permanent stop of running activities. Insight into injury etiology is therefore necessary. Here, the algorithms from the present study could provide epidemiological researchers with new tools to quantify training load with improved validity. To accomplish this, researchers can use the equations in the present manuscript to estimate second-specific approximations of load during running. They can be summed up to calculate the session-specific cumulative load in the Achilles and patellar tendons. Ultimately, researchers may then investigate whether increases in session-specific cumulative loads are associated with injury occurrence. This can be combined with other exposures to investigate how changes in cumulative training load and other parameters (running shoes, cadence, surface, etc.) influence running injury occurrence. 

This could potentially evolve into a wearable technology that can guide runners in terms of quantity and intensity of running activities considering the injury risk.

## 5. Conclusions

The algorithms developed in the current study were always superior to the distance algorithm. Moreover, the best fitting algorithm was consistently superior to the practically feasible algorithm. Therefore, it can be concluded that, when the necessary data is available, the best-fitted algorithm should be used to approximate the peak force and/or impulse in the patellar and Achilles tendon. When only data for the practically feasible algorithm are available, this should be used in favor of the distance algorithm.

## Figures and Tables

**Figure 1 sensors-21-07418-f001:**
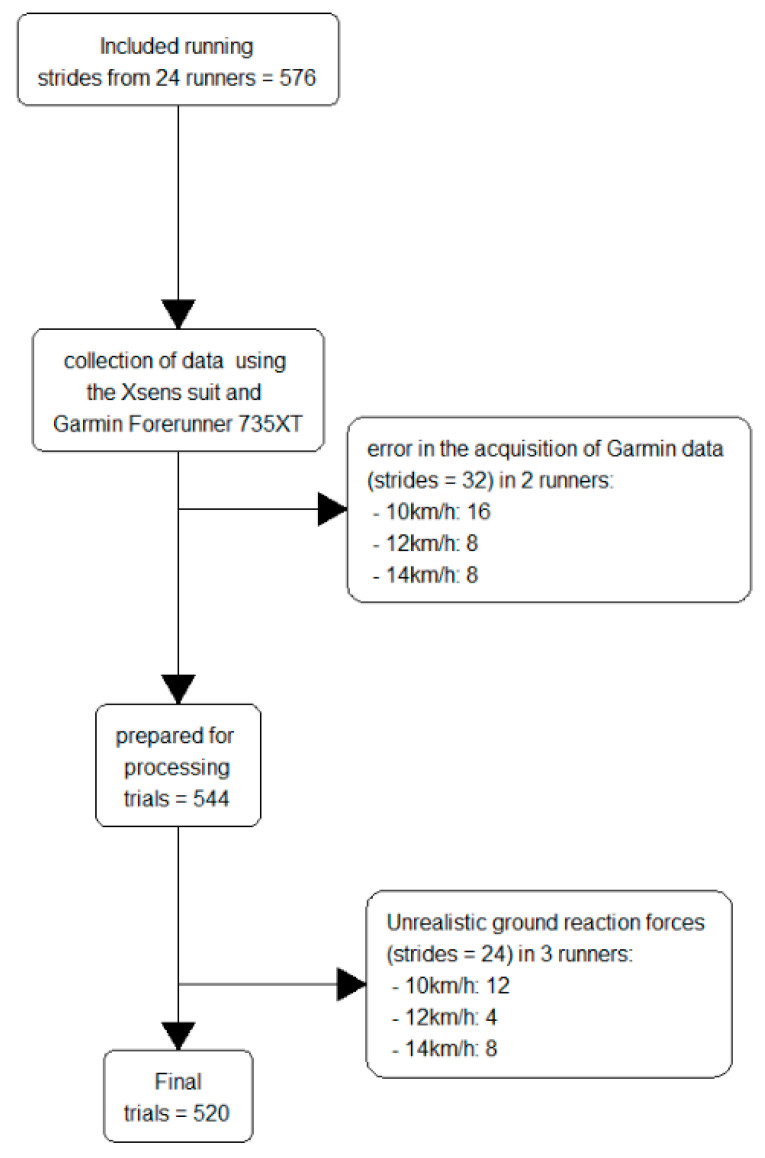
Flow diagram of the process from including runners to processing of data. Twenty-four runners were included for trials of 10, 12, and 14 km/h, respectively, giving 576 strides in total.

**Figure 2 sensors-21-07418-f002:**
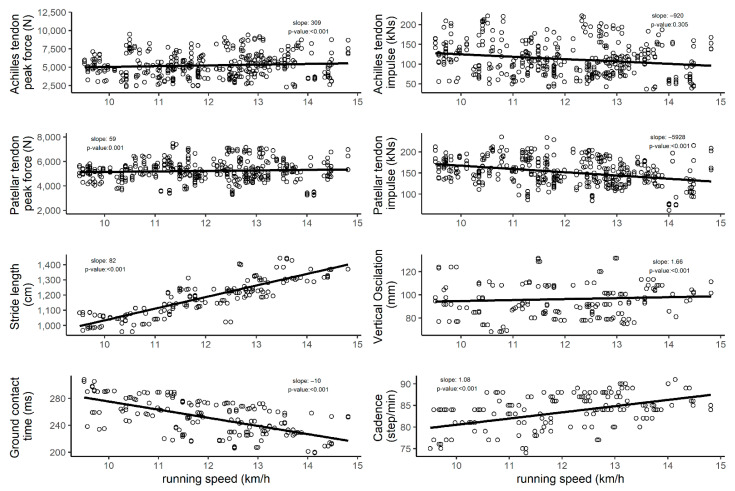
The association between running speed and step-specific tendon force, impulse per stride, stride length, vertical oscillation, ground contact time, and cadence.

**Figure 3 sensors-21-07418-f003:**
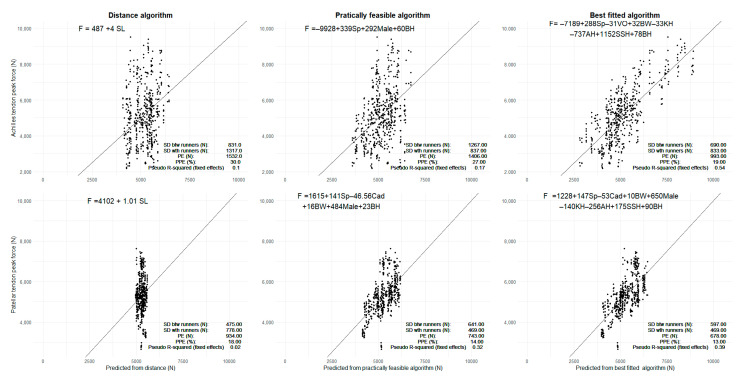
The prediction of peak forces using the different algorithms. The black diagonal line is the perfectly fitted line. The black dots are the AnyBody-estimated force on the *y*-axis and predicted forces on the *x*-axis. SL: Stride length; Sp: Speed; BH: Body height; BW: Bodyweight; KH: Knee height; AH: Ankle height; SSH: Shoe sole height; Cad: Cadence.

**Figure 4 sensors-21-07418-f004:**
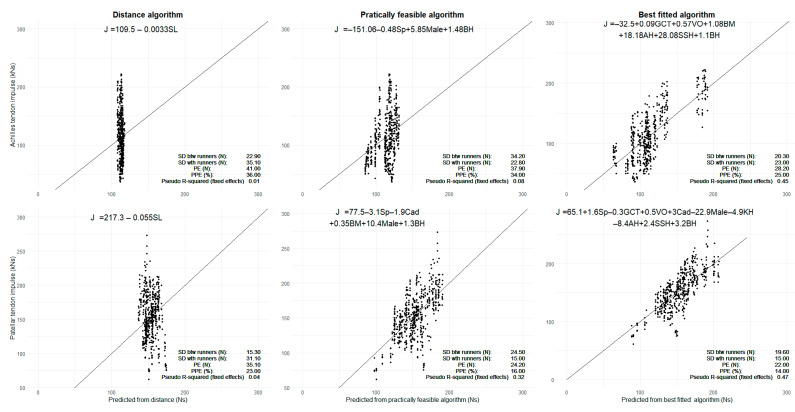
The prediction of impulse forces using the different algorithms. The black diagonal line is the perfectly fitted line. The black dots are the AnyBody-estimated forces on the *y*-axis and predicted forces on the *x*-axis. SL: Stride length; Sp: Speed; BH: Body height; BW: Bodyweight; KH: Knee height; AH: Ankle height; SSH: Shoe sole height; Cad: Cadence.

**Table 1 sensors-21-07418-t001:** The predictive algorithms of the patellar and Achilles tendon peak force during running based on outdoor measurable features.

Structure and Model	Garmin Measurable Variables							Measurable Variables by Runners					Accuracy				
	Intercept (N)	Stride Length (cm)	Speed (km/h)	Ground Contact Time Length (ms)	Vertical Oscilation (mm)	Cadence (Step/min)	Body Mass (kg)	Sex (1 = Male)	Knee Height (cm)	Ankle Height (cm)	Shoe Sole Height (cm)	Body Height (cm)	Standard Deviation between Runners (N)	Standard Deviation within Runners (N)	Prediction Error (N)	Proportion of Prediction Error (%)	Pseudo R-Squared (Fixed Effects)
**Achilles tendon peak load**																	
Distance algorithm	487 [480]	4 [0.33]											831	1317	1532	30	0.1
Practically feasible algorithm	−9928 [6853]		339 ** [29]					292 [662]				60 [39]	1267	837	1406	27	0.17
Best fitted algorithm	−7189 [5530]		288 ** [33]		31 ** [8]		32 * [24]		−33 [104]	−737 ** [214]	−1152 ** [201]	78 [57]	690	833	993	19	0.54
**Patellar tendon peak load**																	
Distance algorithm	4102 [277]	1.01 [0.19]											475	778	934	18	0.02
Practically feasible algorithm	1615 [4731]		141 ** [25]			−46.56 * [18]	16 [22]	484 [373]				23 [31]	641	469	743	14	0.32
Best fitted algorithm	−1228 [5007]		147 ** [25]			−53 ** [19]	10 [23]	650¤ [363]	−140 [93]	−256 [173]	175 [172]	90 ¤ [51]	597	469	678	13	0.39

¤ indicates the *p*-value for the variable to be between 0.05 and 0.1; * indicates the *p*-value for the variable to be less than 0.05; ** indicates the *p*-value for the variable to be less than 0.01; [] indicates the standard error of the estimate.

**Table 2 sensors-21-07418-t002:** The predictive algorithms of the patellar and Achilles tendon impulse load during running based on outdoor measurable features.

Structure and Model	Garmin Measurable Variables							Measurable Variables by Runners					Accuracy				
	Intercept (kN)	Stride Length (cm)	Speed (km/h)	Ground Contact Time Length (ms)	Vertical Oscilation (mm)	Cadence (Step/min)	Body Mass (kg)	Sex (1 = Male)	Knee Height (cm)	Ankle Height (cm)	Shoe Sole Height (cm)	Body Height (cm)	Standard Deviation between Runners (N)	Standard Deviation within Runners (N)	Prediction Error (N)	Proportion of Prediction Error (%)	Pseudo R-Squared (Fixed Effects)
**Achilles tendon loading impulse**																	
Distance algorithm	109.5 [13.07]	−0.0033 [0.009]											22.9	35.1	41.0	36	0.01
Practically feasible algorithm	−151.06 [184]		−0.48 [0.79]					5.85 [17.89]				1.48 [1.05]	34.2	22.8	37.9	34	0.08
Best fitted algorithm	−32.5 [141]			0.09 [0.08]	0.57 * [0.23]		1.08 [0.69]			−18.18 * [6.2]	−28.08 ** [5.91]	1.1 [1.13]	20.3	23	28.2	25	0.45
**Patellar tendon loading impulse**																	
Distance algorithm	217.3 [9.73]	−0.055 [0.01]											31.1	15.3	35.1	23	0.04
Practically feasible algorithm	77.5 [176]		−3.1 [0.83]			−1.9 [0.61]	0.35 [0.85]	10.4 [14]				1.3 [1.19]	24.5	15	24.2	16	0.32
Best fitted algorithm	65.22 [148]		1.62 [1.35]	0.3 * [0.12]	−0.46 ¤ [0.26]	−3 * [1.03]		22.87 * [10.57]	−4.86 [2.81]	−8.44 * [5.6]	2.4 [5.65]	3.16 ** [1.2]	19.6	15	22.0	14	0.47

¤ indicates the *p*-value for the variable to be between 0.05 and 0.1; * indicates the *p*-value for the variable to be less than 0.05; ** indicates the *p*-value for the variable to be less than 0.01; [] indicates the standard error of the estimate.

## Data Availability

The data presented in this study are available on reasonable request from the corresponding author.

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
