# Peer review of "How Precisely Can Easily Accessible Variables Predict Achilles and Patellar Tendon Forces during Running?"

_sensors, 2021, doi:10.3390/s21217418_

Round 1

Reviewer 1 Report

Dear authors, congratulation for this study. Please, find the comments inserted in the main manuscript.

Reviewer 2 Report

The authors have prepared inherently interesting research on the prediction of the patellar and Achilles’ tendon peak force. The focus is related to the analysis of data from Xsens suit and Garmin Forerunner devices, however the adequate description of Xsens device was not found in the manuscript. Therefore, I would suggest adding some necessary information what kind of data is provided from Xsens, especially for the readers, who are not familiar with this kind of devices.

In my opinion the research contains a serious bias. The measurement of the speed should be performed with a specialized device, not with the person riding a bike in front of the runner. Therefore, it is difficult to estimate the exact measurement error in the speed evaluation.

A mixed -effects multiple regression model was used to assess the association between the estimation forces and training load variables. However, authors did not provide the analysis of significance of each parameter on the predictor variable. It is hard to interpret the results without listing in a table the analyzed parameters with estimated values, standard errors, and p-value. Therefore, the predictive equations cannot be interpreted unambiguously.

Round 2

Reviewer 2 Report

The authors have addressed most of my concerns. In my opinion the bias associated with the speed’s measurement should be discussed in the section 4.3 Limitations. Most of the citations are self-citations; the authors should consider removing irrelevant self-citations.

I would suggest changing a few parts of the manuscript in order to increase its readability:

  1. Description of the distance algorithm can be removed. The analysis of the distance algorithm does not contribute anything to the work.
  2. In the section “2.5. Statistics” lines 178-184: I would suggest listing one more time the predictive variables for practically feasible and best fitted algorithms.
  3. The introduction contains a statement that the “injury occurs when the cumulative tendon load exceeds the structure’s capacity to withstand the load”. In my opinion the most valuable impact of this research is a step forward towards estimation of percentage load while running in relation to the maximum load of the tendon. This value could be an individual predictor of the possible tendon injury and could be incorporated into a mobile application. Therefore, I would suggest discussing it in the discussion section.
  4. There are a few English language corrections to be made. I would suggest another English proofreading.

Author Response

Dear reviewer

Thank you for the response to our manuscript. The specific comments have been addressed below and an amended version of the manuscript has been re-submitted. A supplementary marked copy of the manuscript highlights all the changes made to the original submission.

Reviewer 2

The authors have addressed most of my concerns. In my opinion the bias associated with the speed’s measurement should be discussed in the section 4.3 Limitations.

Author Response: This is not clear to us why this is a limitation. We used the measured speed of the runners measured from the watch on the wrist, not the intended speed. However, we have added that controlling for running speed could improve the measurement of running speed.

Author change to manuscript:

The study design is limited to low range running speeds from 10 km/h to 14 km/h and by the lack of precise running speed measurement. It is likely that a plateau exists regarding vertical oscillation, ground contact time, stride length and cadence at faster speeds. In the present study, a linear increase was found for stride length, ground contact time, and cadence, while a slightly progressive increase in vertical oscillation was found. However, studies including higher speed have demonstrated that stride length increases at a lower rate when the speed increases[26]. Consequently, the opposite seems to be the case for cadence, which had progressively higher increases in cadence over time. However, Dorn et al.[26] demonstrated a linear reduction in ground contact time from 12.5-32km/h which may cover the range of speed performed during endurance running. This may indicate that the present study was limited in the speed range and the algorithm should be used with caution for predicting structural forces below 10 km/h and above 14km/h. Moreover, the dataset was rather small and it can be questioned whether the runners included are representative of all types of runners.

Since the data had rather large noise/random variance, the prediction algorithms may display reduced predictive performance. Several improvements have the potential to reduce this noise: Firstly, the Garmin device is not providing stride-to-stride data. Consequently, it was not possible to synchronize the stride-to-stride data from Xsens precisely with the stride-to-stride data from Garmin. Instead, we used second-to-second data from the Garmin device. Secondly, the measures we used are only an average of a series of previous steps, which together with the lack of stride data may lead to imprecise predictions of the tissue loading. Stride-to-stride data without averages of previous steps will improve the fitting of the algorithm. This will ensure an appropriate synchronization between data. Thirdly, another way of reducing the noise could be to change the prediction of ground reaction forces from full body kinematics to measuring ground reaction forces. Finally, control of running speed could be improved by better measurement techniques, and this might reduce random noise and enhance the significance of the findings. Still, the algorithms may prove beneficial in their current form, as the prediction error of the algorithms were at least 100N below the prediction error derived from using the running distance algorithm, which is the commonly used exposure metrics in previous literature[10].

Most of the citations are self-citations; the authors should consider removing irrelevant self-citations.

Author Response: Thanks.

Author change to manuscript: we have removed two citations from the introduction.

I would suggest changing a few parts of the manuscript in order to increase its readability:

  1. Description of the distance algorithm can be removed. The analysis of the distance algorithm does not contribute anything to the work.

Author Response: Comparing the new algorithms to the standard measurement of training load today is of utmost importance if we want to further running-related injury research. As described in this paper (10.26603/ijspt20180931), running distance is the most used measure and therefore we need to have this comparison to evaluate the new algorithms against the standard

  1. In the section “2.5. Statistics” lines 178-184: I would suggest listing one more time the predictive variables for practically feasible and best fitted algorithms.

Author Response: Thanks. It can be difficult to comprehend all the information. We think it’s easier to get an impression of the predictive variables from a visual inspection, we have referred to table 2 and 3 to make this easier.

Author change to manuscript:

For each of the four response variables, three models were obtained. The first model  is based on the “distance” as input, which is the common way in the literature of estimating cumulative load and serves as a control[10]. The second model is the “best fitted”, which considers all available predictors and serves as a benchmark for the predictive performance and when possible, should preferably be used (See table 2 and 3). The third model is the “practically feasible” which considers only predictors, which may be assessed in a valid manner by runners themselves.

  1. The introduction contains a statement that the “injury occurs when the cumulative tendon load exceeds the structure’s capacity to withstand the load”. In my opinion the most valuable impact of this research is a step forward towards estimation of percentage load while running in relation to the maximum load of the tendon. This value could be an individual predictor of the possible tendon injury and could be incorporated into a mobile application. Therefore, I would suggest discussing it in the discussion section.

Author Response: Thank you. We have added a perspective section in the discussion around this.

Author change to manuscript:

4.4. perspectives

Running injuries are commonly affecting runners, which can lead to a temporal, and even a permanent, stop of running Running injuries are commonly affecting runners and can lead to a temporary or permanent stop of running activities. Insight into injury etiology is therefore necessary. Here, the algorithms from the present study could provide epidemiological researchers new tools to quantify training load with improved validity. To accomplish this, researchers can use the equations in the present manuscript to estimate second-specific approximations of load during running. They can be summed up to calculate the session-specific cumulative load in the Achilles and patellar tendons. Ultimately, researchers may then investigate whether increases in session-specific cumulative loads are associated with injury occurrence. This can be combined with other exposures to investigate how changes in cumulative training load and other parameters (running shoes, cadence, surface etc.) influence running injury occurrence.

                           This could potentially evolve into a wearable technology which has the ability to guide runners in terms of quantity and intensity of running acitivities considering the injury risk.

  1. There are a few English language corrections to be made. I would suggest another English proofreading.

Author Response: Thanks. We have been through the manuscript for English language corrections.